# Miniscrew-Assisted Rapid Palatal Expansion: A Scoping Review of Influencing Factors, Side Effects, and Soft Tissue Alterations

**DOI:** 10.3390/biomedicines12112438

**Published:** 2024-10-24

**Authors:** Anca Labunet, Cristina Iosif, Andreea Kui, Alexandra Vigu, Sorina Sava

**Affiliations:** 1Dental Materials and Ergonomics Discipline, Faculty of Dental Medicine, “Iuliu Hatieganu” Medicine and Pharmacy University, Avram Iancu 31 Street, 400083 Cluj-Napoca, Romania; labunet@yahoo.com (A.L.); alexandravigu@elearn.umfcluj.ro (A.V.); savasorina@elearn.umfcluj.ro (S.S.); 2Prosthetic Dentistry Discipline, Faculty of Dental Medicine, “Iuliu Hatieganu” Medicine and Pharmacy University, Clinicilor 32 Street, 400006 Cluj-Napoca, Romania

**Keywords:** MARPE, maxillary expansion, orthodontic techniques, microimplant, bone-borne maxillary expansion

## Abstract

**Background**: Miniscrew-assisted rapid palatal expansion (MARPE) has gained attention as an effective alternative to traditional rapid palatal expansion, particularly in adult patients. This scoping review synthesizes recent evidence to assess the clinical efficacy and safety of MARPE, addressing a gap in comprehensive, up-to-date analyses in this area. **Objective**: To present the recent assessments concerning MARPE influencing factors, side effects, soft tissue alterations, and airway changes, focusing on comparisons with conventional devices. **Methods**: Using PRISMA guidelines, we conducted a search of the literature published in 2018–2023 using Medline, Scopus, and Embase databases. This review focused on randomized controlled trials, cohort studies, and other reviews that evaluated the outcomes of MARPE. **Results**: Our analysis included 75 studies and revealed that MARPE significantly improves suture expansion with fewer dental and skeletal side effects compared to traditional methods. The technique shows high efficacy in subjects up to 25 years of age, with reduced incidence of complications and improved stability of expansion. **Conclusions**: The results support MARPE as a viable and superior alternative for maxillary expansion in late adolescents and adults. Given its advantages over traditional methods, MARPE should be considered a standard procedure in orthodontic treatment plans. Future research should focus on long-term outcomes and optimization of patient-specific treatment protocols.

## 1. Introduction

Maxillary transverse discrepancy is a relatively common condition in orthodontic patients, often leading to a unilateral or bilateral posterior crossbite. It is caused by genetic or environmental factors and is often associated with dental crowding [1]. This condition may lead to complications such as canine impaction, crossbite, Class II and III malocclusions, temporomandibular joint disorders, and obstructive sleep apnea. Treatment typically involves maxillary disjunction initiated pre-puberty [1,2]. Rapid maxillary expansion [RME/RPE/HYRAX] devices are commonly used [3] to separate the midpalatal suture. The device applies bilateral forces from the expansion screw through the first upper molars and premolars to the palatal bone, indirectly causing the separation of the midpalatal suture, when not fully formed. However, in late adolescents and adults, these techniques are less effective due to the maturation of the median palatal suture, which reduces efficacy and leads to side effects such as dentoalveolar compensation [3] and undesirable dental and periodontal effects [4,5,6].

Surgically assisted rapid palatal expansion [SARPE] was traditionally the alternative for these patients, but the associated risks and patient discomfort often reduce compliance. Recently, miniscrew-assisted rapid palatal expansion [MARPE] has emerged as a new treatment option for post-pubertal patients [7,8]. MARPE is either a tooth–bone-borne or solely bone-borne device that includes a rigid element connected to miniscrews inserted into the palate. This configuration delivers the expansion force directly to the basal bone of the maxilla, enhancing skeletal expansion.

MARPE appears to lower the risk of dentoalveolar compensations and unwanted effects [5,6]. Compared to SARPE, MARPE is a less complicated technique with few impacts on patient-reported outcomes and lower costs [3,5]. However, SARPE is a previously proven measure that guarantees a correction of transverse maxillary deficiency, with some surgical techniques having lower side effects and no differences in the symmetry and amount of expansion [9]. In MARPE, integrating a digital workflow into the traditional analog protocol has proven feasible [10].

As studies focus mainly on one of the several aspects of the MARPE device’s efficacy and consequences and there was no review found integrating different outcomes of the appliance, we believe this integrated research is essential to orthodontists. This scoping review aims to offer an integrative outlook on MARPE studies and reviews published in the last 5 years. The objective is to provide an outlook on factors influencing treatment, soft tissue changes, effects on airways and breathing, and side effects and offer comparisons between devices.

## 2. Materials and Methods

This scoping review was conducted following the PRISMA guidelines [Preferred Reporting Items for Scoping Reviews and Meta-Analyses] [11]. In addition, the research question was defined through the PICOT format [population, intervention, comparison, outcomes, and time] [12]: P: Individuals requiring maxillary expansion, I: use of bone-borne or hybrid microimplants maxillary expanders, C: use of MARPE, O: factors influencing treatment outcome, soft tissue changes following expansion, effects on airways, and sleep apnea as well as side effects of MARPE and comparisons with other expander systems, T: a contemporary overview of studies from the past 5 years.

### 2.1. Information Sources and Search Strategy

The search was initiated on 1 February 2024 and conducted through 20 July 2024 by two reviewers [AL and AK] using the following bibliographic databases: Medline [PubMed], Scopus, and Embase. We established 3 search concepts (Table 1) and, based on these, developed keywords and search items, including MeSH terms, for use across all three databases. The exact search combinations for each database are detailed in Table 2. In addition to the database searches, a manual search was performed, and references from various studies were reviewed to identify additional relevant and eligible studies. No automation tools were used.

### 2.2. Eligibility Criteria

#### 2.2.1. Inclusion Criteria

Randomized controlled trials, cohort studies, case reports, case-control studies, and studies involving patients who had undergone maxillary expansion by means of any microimplant system.Review articles and meta-analyses that provided comprehensive overviews or evaluations of the topic.Articles focusing on different types of maxillary expanders that evaluated factors influencing treatment outcomes, soft tissue changes following expansion, effects on airways and sleep apnea, and side effects of MARPE and offered comparisons with other expander systems.Studies published in English and completed between 2018 and 2023.

#### 2.2.2. Exclusion Criteria

Studies involving the use of conventional maxillary expanders, without the use of microimplants.Studies performed on animal or artificial bone and in vitro studies.Studies on computer models simulating maxillary expansion or digital planning.Studies on a certain/new type of maxillary expander specifically or workflow presentation studies.Studies that did not report any of the key outcomes of interest for the review.Articles published in languages other than English and those older than 5 years.

For data extraction, a standardized form was used and recorded in an Excel table (v.15.17—Microsoft, Redmond, WA, USA). The extracted information included the following: bibliographic details [authors, title, year of publication, journal], study design and methodology, sample size and demographics, types of maxillary expanders used, outcomes [such as mechanical strength, aesthetic features, adhesion properties, etc.], key findings, and conclusions. The standardized form was pilot-tested by team members using five randomly selected studies.

The two reviewers [AL and AK] extracted data to ensure consistency and reduce potential bias. Any discrepancies between the two reviewers were resolved through discussion or by consulting a third reviewer [AV]. Consequently, we evaluated the quality of the articles for this scoping review and found articles that matched the quality assessment. The reviewers developed a template to extract key information from the studies, including the study purpose, sample, groups, and results relevant to the review’s objectives. Data extraction was initially conducted by two reviewers and then reviewed for accuracy and completeness by two additional investigators [AV and SS]. The risk of bias was considered in regard to the study design, funding, and data analysis. Thus, considering the subject at hand, researchers focused on bias in the selection of participants and appliances, blinding of data collection and analysis, and control of confounders. Risk of bias was evaluated by two researchers [AV and CI]. Risk of bias was considered minimal for articles included in this scoping review.

## 3. Results

### Data Collection

A total of 125 articles were identified after applying the search strategy (as outlined in Table 1 and Table 2). Following the removal of duplicates and articles unrelated to the topic, 110 records were retained for screening, all of them matching the quality criteria. Quality assessment for articles included sample size, use of control groups whenever possible, discussion of findings relative to the existing literature, evaluation of the degree of generalization from the research sample to the population, applicability of evidence, and collection and analysis of data. In the first phase, articles were selected based on their titles and abstracts, focusing on their relevance to the study question. This screening process resulted in 87 articles, of which 79 were further assessed for eligibility. Any disagreements were resolved through discussion and consultation with a third and fourth researcher. Ultimately, 75 publications were included in this review. The selection process, along with the inclusion decision, is shown in Figure 1, the PRISMA flow diagram, according to the inclusion and exclusion criteria presented above in Section 2.2.

Most studies focused on factors influencing MARPE outcomes (25 studies); this was followed by those focused on adverse effects (18 studies) and airway changes (17 studies). Just eight studies researched changes in soft tissue and seven studies compared different types of appliances. Some articles obtained focused on several aspects listed above but were included in one category for clarification purposes. Twenty-eight of the articles included in the study were reviews published within the researched time frame. Some heterogeneity between results of similar studies was revised through subgroup analyses indicating the potential outcome modifiers. Findings showed that participant age and differences in study design may have led to opposing conclusions. The number of participants in studies also provided higher confidence in the results. As there was no blinding possible in the studies, some operator bias may have been present in all study outcomes.

## 4. Discussion

This review focused on five distinct areas of research: aspects leading to MARPE success; soft tissue alteration following bone-borne or hybrid disjunction; influence on airways and sleep quality; side effects such as root resorption, loss of bone tissue, and relapse; and, finally, comparisons with other types of disjunction devices.

### 4.1. Factors Influencing MARPE

MARPE is more predictable and achieves a more substantial suture expansion than the Hyrax expander, with fewer side effects [13]. Several studies place MARPE efficacy around 80% [14].

Several factors for MARPE success have been considered by researchers. Age, palatal suture maturation, and sex show the most significance. However, studies sometimes have opposite conclusions.

No difference in MARPE efficacy between male and female patients was observed. Age, palate length, and midpalatal suture maturation stage can predict the midpalatal suture expansion by MARPE in young adults [15,16,17]. There was a statistically significant association between older age and suture nonseparation in males but not in females. In suture-separated subjects, there was a statistically significant trend toward a low amount of suture separation in older age subgroups for both sexes [18]. As age increased, MARPE success and the skeletal effects of maxillary expansion decreased [14,19]. All instances of MARPE failure were observed in patients at late stages of suture maturation. However, there was a certain degree of efficacy of MARPE even in older patients: although more patients exhibited more significant skeletal and dental changes, both groups showed a similar ratio of skeletal-to-dental transverse changes [20].

A significant correlation was identified between post-expansion changes and the thickness of the palatal cortical bone, as well as the inclination of the palatal plane. Thicker cortical bone in the palate and/or flatter palatal planes were associated with improved stability [21].

Age showed a negative correlation with bone expansion, alveolar expansion, and alveolar change [22,23]. Additionally, a negative correlation was observed between MPSM and variations in the nasal cavity, bone expansion, and alveolar change. MARPE could more easily expand the midpalatal suture in patients under 20 years of age compared to those 20 years and older [22,24,25].

By comparing dentoskeletal and periodontal changes after MARPE in patients aged 18–29 and 30–45 years, Naveda reported a midpalatal suture opening success rate of 100% and 81%, respectively. No significant differences were found between the groups in terms of increases in maxillary and dental arch widths, and the buccal tipping of anchorage teeth was observed to be similar in both groups. After expansion, buccal bone thickness of the posterior teeth decreased, while palatal bone thickness increased, with no differences noted between the two age groups [26]. Adult patients in stage D of maxillary suture ossification were more prone to dentoskeletal changes after MARPE therapy compared to those in stage E. 51, although suture width increased after MARPE even in adult patients [5,27].

These findings were contradicted by one study that found no significant differences between separated and non-separated midpalatal sutures in terms of age, vertical and horizontal skeletal relationships, or palate length. However, the zygomaticomaxillary, pterygomaxillary, midpalatal, and transverse palatine sutures displayed significantly greater widths in the separation group compared to the non-separation group, even within this study [28].

Thinner midpalatal suture bone in areas 18 mm and 21 mm behind the incisor foramen, along with thicker palatal bone and nasal cortical bone at anterior microimplant positions, were associated with more effective MARPE outcomes [29]. As older age, thin palatal bone, and higher stage of maturation can influence MARPE, the corticopuncture technique appears to have a positive impact [30].

Implant placement may be a predictor for MARPE success. An analysis of 223 CBCTs [Cone Beam Computed Tomography] showed considerable variability in palatal bone thickness both among patients and across different regions, showing that a prior evaluation of palatal bone thickness is crucial for identifying the optimal sites and angles for mini-implant placement [31]. There is no difference between anterior or posterior mini-implants when analyzing a displacement pattern [32].

A team led by Di Leonardo studied torque values for mini-implants, showing that torque values between 10 and 20 Ncm are acceptable and will produce favorable effects [33].

By evaluating the effects of bicortical engagement in mini-implants associated with maxillary skeletal expanders on the opening of pterygopalatine sutures and analyzing the post-expansion skeletal changes, one study found that bicortical microimplant anchorage is crucial for the opening of the pterygopalatine suture [34]; this was contradicted by one study where bicortical engagement showed no difference in outcomes [20].

One study compared two different MARPE systems: one using the maxillary first premolars, maxillary first molars, and four microimplants as anchors and the other utilizing only the maxillary first molars and microimplants as anchors. The second MARPE system demonstrated greater posterior expansion at both the alveolar and basal bone levels, with an almost parallel split. Both systems exhibited a pyramidal expansion pattern in the coronal view [35].

### 4.2. Soft Tissue Changes

Some soft tissue modifications are a direct consequence of bone widening and may be present both extra- and intraorally.

There were significant changes in the paranasal, upper lip, and both cheeks following expansion using MARPE, with greater magnitude in the cheek areas, stable after 1 year [36,37]. The nose tended to widen and move forward and downward. The post-treatment nasal volume could also exhibit a significant increase relative to the initial volume [38]. These findings were contradicted by some studies that did not find any significant changes in soft tissues [39,40].

Statistically significant differences in the soft tissue parameters were observed, which included an increased H-angle, increased soft tissue subnasal to the H-line, and a decreased soft palate surface area after MARPE [41].

Chen analyzed the interdental papilla height of maxillary central incisors after MARPE. Eighteen months later, 18% of patients exhibited mild recession of papilla height of the maxillary central incisors, with prognostic factors overlapping and a smaller crown width-to-length ratio of maxillary central incisors [42].

Some palatal mucosa inflammation is usually present in expansion treatments. The type of palatal expander influences the degree of inflammation, with the severity of hyperplasia being more pronounced in the case of MARPE than in the case of RPE usage. The incidence and severity of the inflammatory reaction in the palatal mucosa were variable in patients with the same distance between the expander and the palatal or gingival mucosa [43]. Although swelling was present in all cases, there were no cases of necrosis of the palatal mucosa [44].

### 4.3. Sleep Apnea and Airways

Increased width of the maxillary and nasal floor improves airway function and reduces sleep apnea, as shown by most studies. MARPE treatment produced an increased objective and subjective airway improvement that continued to remain stable in the long term post-expansion [45], confirmed by several review studies [46,47,48,49] and other research [50,51,52,53]. Clinical results of MARPE seemed to be superior to conventional RPE or SARPE [54]. In addition, MARPE had better results than RPE in the posterior area [52]. A meta-analysis led by Liu revealed that MARPE can increase nasal cavity width, nasal cavity volume, nasopharyngeal volume, and oropharyngeal volume for nongrowing patients but has no significant effect on hypopharyngeal volume [55], as shown in [52]. In the late stages of ossification, D and E, there were MARPE-associated improvements in the upper airway [56].

Similarly, researchers led by Yoon showed that MARPE treatment reduces the severity of obstructive sleep apnea, refractory nasal obstruction, and daytime somnolence and increases the percentage of REM sleep in adult patients with a narrow maxilla and nasal floor [57].

MARPE enhanced anatomical characteristics, such as the total volume of the upper airway and the minimum cross-sectional area. It also produced favorable changes in aerodynamic characteristics, including reduced resistance, decreased velocity, and lower minimum wall shear stress [58]. Researchers observed important daytime sleepiness and obstructive sleep apnea-related quality of life improvement, as well as higher oxygen saturation and reduced snoring duration [59].

A significant increase in total area and minimal section at the level of the nasopharynx and oropharynx was observed in cases treated with bone-borne maxillary expansion. Regarding the comparison of MARPE and bone-borne maxillary expansion treatments, no differences were found in the total airway volume and minimal section in the upper airway except for the minimum cross-section of the nasal cavity, which increased for MARPE and decreased for bone-borne maxillary expansion [60].

Contrary to this, two reviews found that the short-term airway volumetric changes secondary to MARPE were not significant [49,61].

### 4.4. Side Effects

MARPE is generally a well-tolerated expansion treatment. At the beginning of the expansion process, patients may experience a temporary decline in Oral Health-Related Quality of Life [OHRQoL] and moderate pain. However, OHRQoL improves, and pain becomes very mild as the treatment progresses [42].

Some side effects are generally present with MARPE treatment, although reduced in severity compared with SARPE or conventional RPE [62,63]. A team led by Haas found that complications were significantly associated with age. Therefore, a careful expansion protocol appears to be beneficial in preventing unfavorable outcomes, particularly in adult patients [64]. By comparing RPE and MARPE, one study showed significant triangular basal bone expansion and skeletal relapse during the consolidation phase. However, with the same amount of expansion, the MARPE group showed a smaller reduction in skeletal, dentoalveolar, and periodontal variables after consolidation. Reinforcing RPE with miniscrews helped maintain basal bone stability during the consolidation period [65]. A greater efficacy of MARPE was noted by several studies [65,66]; however, a small amount of relapse was observed 1 year after expansion [67,68]. In the same context, MARPE caused tooth inclination and a decrease in alveolar height, but it was less significant than with conventional RPE, as shown in a review [67].

Minor dentoalveolar changes were observed following MARPE, such as a reduction in buccal bone thickness, which was not clinically detectable [4,69]. Contrary to this, one study found a significant reduction in buccal bone thickness and a significant buccal inclination of first molars. A greater maxillary transverse increase was associated with a larger intermolar width increase, as well as more significant buccal bone loss at the mesiobuccal roots of the maxillary first molars [14]. Most maxillary expansion in adolescents resulted from skeletal expansion, preserving the alveolar bone and causing only minimal buccal dental tipping [69].

A reduction in pulp blood flow was observed during rapid maxillary expansion with most expansion appliances; however, the decrease was less pronounced in the MARPE group compared to the RME group two weeks after expansion. Nevertheless, regardless of the history of trauma, most teeth in all groups retained pulp sensitivity at all time intervals [70]. By measuring the optic nerve sheath diameter under ultrasonography guidance, researchers evaluated possible intracranial pressure changes caused by screw activations during active the microimplant-assisted rapid palatal expansion of post-pubertal individuals and found no changes in intracranial pressure [71].

However, when listing negative side effects, there is evidence that teeth supporting the expansion appliance in MARPE patients show torque increase compared to cases treated with BAME [72]. Also, an initially asymmetric position of the mid-palatal suture generally results in asymmetric skeletal expansion during treatment [73]. Some studies that focused on dental and periodontal side effects and soft tissue effects concluded that MARPE may cause dental and periodontal side effects and affect peri-oral soft tissues [5,49,68]. A decrease in bone density was observed after the retention period compared to the pre-expansion stage. Additionally, most adult patients showed incomplete repair of the midpalatal suture 16 months following MARPE [74].

A study led by Cantarella aimed to evaluate midfacial skeletal changes in the coronal plane and the implications of circummaxillary sutures and localize the center of rotation for the zygomaticomaxillary complex after therapy with a bone-anchored maxillary expander. A significant lateral displacement of the zygomaticomaxillary complex occurred in late adolescent patients and the zygomatic bone tended to rotate outward along with the maxilla, with a common center of rotation located near the superior aspect of the frontozygomatic suture. However, tipping of the molars was negligible during treatment [75].

By investigating the immediate and long-term effects of conventional and MARPE appliances on root resorption, researchers found that long-term root resorption was similar between the RPE, MARPE, and control groups. Molar inclination showed a significant negative association with the length of the mesiobuccal root of the first molar [76]. One review found moderate evidence of reduced root resorption when compared with other palatal disjunction methods [50].

After MARPE, a certain degree of relapse was typically observed. According to research led by Tang, there was a 5.75% reduction in the total expansion of nasal width and a 19.75% decrease in the lateral pterygoid plate in young adults [21].

### 4.5. Comparisons

Several studies aimed to compare the effects and side effects of various expansion appliances, reaching diverse findings. Bone-borne expanders showed more effective orthopedic effects and fewer dentoalveolar side effects compared to the hyrax expanders [77,78]. Tissue-borne MARPE achieved the same expansion efficiency as tooth-borne MARPE. However, tooth-borne MARPE resulted in more dentoalveolar side effects, including greater buccal tipping, root resorption, and alveolar bone loss [79].

By comparing the effects of a hybrid miniscrew-supported expander versus a conventional Hyrax expander in growing patients, one study showed that the hybrid group was more effective in increasing nasal cavity width, maxillary width, and buccal alveolar crest width. They observed no differences in intermolar, interpremolar, or intercanine widths or arch length, perimeter, size, or shape [80,81].

Alsayegh aimed to research arch parameters and dentoalveolar changes from pretreatment to posttreatment by comparing MARPE, Periodontally Accelerated Osteogenic Orthodontics [PAOO], and Damon self-ligating bracket therapies. The greatest increase in inter-molar width was demonstrated by MARPE, followed by PAOO and Damon. Moreover, MARPE was the only group to produce a significant increase in palatal vault area [82].

When comparing micro-implant placement influence on the efficacy of MARPE treatments, one study showed that 2-rear-bicortical penetrating mini-implants were essential for skeletal expansion [83].

MARPE allows for more predictable and significant skeletal expansion, with reduced buccal tipping and less alveolar height loss on the anchorage teeth. As a result, MARPE is a superior alternative for patients with skeletal maxillary deficiency during the post-pubertal growth spurt stage [84].

### 4.6. Limitations of This Research

We identified several limitations to our review. Firstly, the included studies varied considerably in design, sample size, and methodology, which may have affected the comparability of the results. Differences in how studies define and measure outcomes can lead to heterogeneity, making it difficult to draw generalized conclusions. In addition, our review primarily included studies focusing on specific age groups, such as late adolescents and adults, which may limit the applicability of the results to other age groups, such as younger children. Also, demographic factors such as race and ethnicity, which may influence craniofacial anatomy and response to treatment, may not be adequately represented.

Many studies of MARPE may have short follow-up periods that do not adequately capture long-term outcomes, stability of enlargement, or late-onset complications. The lack of long-term data may limit the ability to fully assess the efficacy and safety of MARPE over time. In addition, the use of different MARPE devices in different trials may lead to variability in results. Differences in the design and use of MARPE devices, such as the number of miniscrews used and their placement, may affect the results and their interpretation.

Comparisons with established treatment plans focusing on SARPE and conventional RPE were not thoroughly highlighted in this scoping review as the inclusion and exclusion criteria applied, according to our objective, did not include such research.

Some of the included trials had small sample sizes, lacked randomization, or did not include controls. This may have affected the strength of the evidence and the reviews’ conclusions. In addition, our overview was limited to studies published in the English language, which may have excluded relevant results published in other languages.

### 4.7. Clinical Implications

Miniscrew-assisted rapid palatal expansion (MARPE) has demonstrated increased predictability and efficacy in achieving significant maxillary expansion, particularly in non-growing patients, including adults, where traditional methods are often ineffective due to suture ossification. This suggests that MARPE may significantly alter clinical decision-making in this population. In addition, patient age emerges as a critical factor; younger patients tend to respond more favorably to MARPE, although its efficacy in older patients still exceeds that of traditional methods. The role of anatomical factors is also critical, highlighting the need for detailed diagnostic imaging such as CBCT to tailor treatments to individual anatomical conditions and ensure the success of expansion.

In addition, MARPE was associated with long-term stability and reduced complications such as tooth tipping, periodontal compromise, and root resorption, making it a viable option for adult orthodontic treatment. These findings may expand the scope of MARPE beyond traditional orthodontics to include multidisciplinary care with ENT specialists and sleep physicians, especially if improvements in nasal breathing or the alleviation of obstructive sleep apnea symptoms are observed. Continued technological advances in MARPE devices will require ongoing education and specialized training for orthodontists to effectively manage the complexities of the treatment. The robust evidence supporting MARPE could also lead to updates in clinical practice guidelines, advocating its use in specific cases and potentially guiding future research to explore long-term outcomes and compare different MARPE techniques.

This scoping review was conducted in accordance with PRISMA guidelines for systematic reviews and PRISMA for abstracts. It followed an a priori established protocol, registered on PROTOCOLS.IO (dpcz5ix6). A comprehensive and specific search strategy was used, built to ensure a maximum number of search results while also maintaining high specificity. Confounding factors were partially controlled by using detailed inclusion and exclusion criteria. This scoping review represents a detailed overview of the most recent studies available in the literature while highlighting various methodology aspects to be considered in further study protocols.

## 5. Conclusions

This scoping review consolidated recent evidence to critically evaluate the efficacy and safety of miniscrew-assisted rapid palatal expansion (MARPE). Our analysis of the literature highlights MARPE as a significantly effective treatment modality for maxillary expansion in post-pubertal patients where traditional methods may fail due to ossification of the midpalatal suture.

Key findings indicate that MARPE achieves greater suture expansion with greater predictability and fewer side effects compared to traditional rapid palatal expanders such as the Hyrax device. This is particularly evident in patients up to 25 years of age, where the success rate of suture separation remains high. Importantly, factors such as age, palatal suture maturation, and the biomechanical properties of the palatal bone play a crucial role in influencing the outcome of the expansion process. MARPE customization, when combined with precise miniscrew placement and consideration of individual anatomical variations, increases the predictability of successful outcomes.

Some soft tissue modifications, such as papillary recession, palatal temporary inflammation, and widening at the nasal and cheek areas are direct consequences of bone widening.

MARPE increases in maxillary and nasal floor width improve airway function and reduce sleep apnea, as shown by most studies, thus creating objective and subjective airway improvement that continues to remain stable in the long term post-expansion.

Side effects in MARPE treatments are similar to previously established methods, such as SARPE and conventional RPE, and appear more frequently in older patients. They include vestibular bone reduction, increased torque of supporting teeth, temporary reduction in pulp blood flow, parodontal issues, incomplete repair of the midpalatal suture, long-term root resorption, and relapse.

Within the limitations of this scoping review, we recommend considering MARPE as a standard part of orthodontic treatment plans for appropriate candidates. Future research should aim to further refine patient selection criteria and optimize treatment protocols to maximize the therapeutic benefits of MARPE. In addition, long-term follow-up studies are needed to assess the stability of results and better understand the physiological and biological responses to this treatment.

## Figures and Tables

**Figure 1 biomedicines-12-02438-f001:**
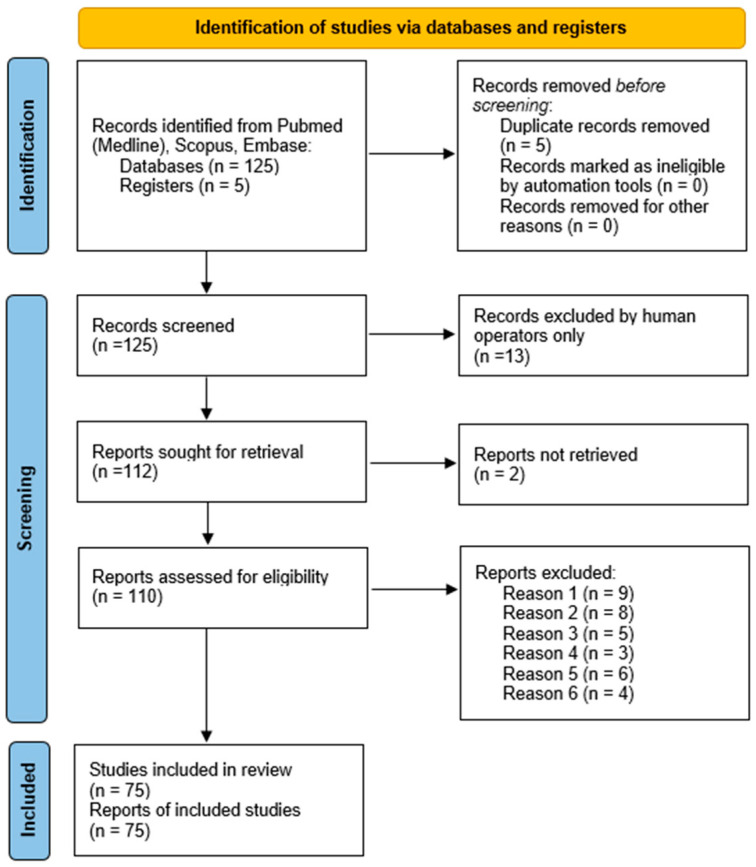
PRISMA flow diagram.

**Table 1 biomedicines-12-02438-t001:** Search concepts.

Concept	Keywords and MeSH Terms
Maxillary expander	“maxillary expan*” [tw] OR “palatal expan*”
MARPE	“MARPE” [tw] OR “Miniscrew assisted rapid palatal expan*” [tw] OR “Microimplant assisted rapid palatal expan*” [tw] OR “Microimplant-assisted rapid palatal expan*” [tw]
Bone-borne expander	“Bone-borne expan*”

**Table 2 biomedicines-12-02438-t002:** Search combinations per database.

Database	Search Terms and Combinations
PubMed Embase Scopus	“maxillary expan*” [tw] OR “palatal expan*” OR “MARPE” [tw] OR “Miniscrew assisted rapid palatal expan*” [tw] OR “Microimplant assisted rapid palatal expan*” [tw] OR “Microimplant-assisted rapid palatal expan*” [tw] OR ”Bone-borne expan*”

## Data Availability

No new data was created or analyzed in this study.

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
