# Peer review of "Miniscrew-Assisted Rapid Palatal Expansion: A Scoping Review of Influencing Factors, Side Effects, and Soft Tissue Alterations"

_biomedicines, 2024, doi:10.3390/biomedicines12112438_

Round 1

Reviewer 1 Report

Comments and Suggestions for Authors

The study design is unclear. Is this article a “scoping review,” as indicated by the title, a “systematic review” as indicated in line 12, or a “literature review” as in line 62?

The manuscript lacks scientific format and contains several mistakes in grammar and language structures.

In line 11, consider changing “non-growing patients” to “adults

In line 13, “To assess the influencing factors for successfully” is not accurate. This review article did not assess but presented a previously published assessment. The word “successfully” is wrong.

In line 14, the findings of “side effects, soft tissue alterations and airway changes” are not presented.

In line 52, “MARPE is a simpler technique with fewer impacts” can be corrected to “MARPE is a simple technique with few impacts”.

In line 111 “we evaluated the quality of the articles for this integrative review”: the outcomes for this evaluation are not mentioned in the results.

In line 116, “The risk of bias analysis was based on three dimensions”: ”: the outcomes for this risk analysis are not mentioned in the results.

In lines 117-119, the statement is unclear and requires clarification.

Line 130- 132, “The selection process, along with the inclusion decision, is shown in Figure 1, the PRISMA flow diagram”. This diagram contains undefined items such as reasons (1-5)

Line 137, “also including research older than 5 years. Some heterogeneity between results of similar studies”. This contradicts the inclusion criteria.

Most of the discussion section is related to the results section. The discussion section should interpret the study findings and provide explanations and comparisons to other studies.

The conclusions section should provide the main ideas or concepts from the study findings.

This article requires professional revision of language and structures.

Comments on the Quality of English Language

This article requires professional revision of language and structures.

Author Response

Dear Reviewer 1,

Thank you for your valuable feedback and suggestions. We have considered all your comments and we have performed several substantial changes to our manuscript accordingly.

Please find below the point-by-point answer to each comment:

Comment 1: The study design is unclear. Is this article a “scoping review,” as indicated by the title, a “systematic review” as indicated in line 12, or a “literature review” as in line 62?

Response 1: This study is a scoping review, corrected lines 12, 64.

Comment 2: The manuscript lacks scientific format and contains several mistakes in grammar and language structures.

Response 2: Corrections and updates have been made throughout the manuscript.

Comment 3: In line 11, consider changing “non-growing patients” to “adults.

Response 3: Item corrected, line 11.

Comment 4: In line 13, “To assess the influencing factors for successfully” is not accurate. This review article did not assess but presented a previously published assessment. The word “successfully” is wrong.

Response 4: Lines 13, 14 have been corrected.

Comment 5: In line 14, the findings of “side effects, soft tissue alterations and airway changes” are not presented.

Response 5: Please find Discussion section subchapter 4.1-4.5.

Comment 6: In line 52, “MARPE is a simpler technique with fewer impacts” can be corrected to “MARPE is a simple technique with few impacts”.

Response 6: Corrected, lines 51-52.

Comment 7: In line 111 “we evaluated the quality of the articles for this integrative review”: the outcomes for this evaluation are not mentioned in the results.

Response 7: Changes have been made to include information on the topic, see lines 113,114, 131-134.

Comment 8: In line 116, “The risk of bias analysis was based on three dimensions”: ”: the outcomes for this risk analysis are not mentioned in the results.

Response 8: Lines 118-123 updated.

Comment 9: In lines 117-119, the statement is unclear and requires clarification.

Response 9: lines 111-123 updated.

Comment 10: Line 130- 132, “The selection process, along with the inclusion decision, is shown in Figure 1, the PRISMA flow diagram”. This diagram contains undefined items such as reasons (1-5).

Response 10: Clarification included, lines 138-141.

Comment 11: Line 137, “also including research older than 5 years. Some heterogeneity between results of similar studies”. This contradicts the inclusion criteria.

Reespons 11: Clarification in lines 146, 147.

Comment 12: Most of the discussion section is related to the results section. The discussion section should interpret the study findings and provide explanations and comparisons to other studies.

Response 12: please find updates Discussion section. Studies and reviews identified as explained in the Results section are discussed and compared.

Comment 13: The conclusions section should provide the main ideas or concepts from the study findings.

Response 13: Conclusions section has been improved as you suggested.

Comment 14: This article requires professional revision of language and structures.

Response 14: English proof reading has been performed, mistakes have been corrected, as suggested.

Kind regards,

 Dr. Andreea Kui

Reviewer 2 Report

Comments and Suggestions for Authors

The methodology section of this literature review appears to be well-structured and adheres broadly to the PRISMA guidelines. However, there are some critical aspects that warrant discussion, particularly concerning the scope of the literature search and the potential limitations arising from the defined 5-year review period. In particular, the investigations into SARPE are underrepresented due to the time limitation. A current systematic review could be found here: Barone, S., Bennardo, F., Salviati, M. et al. Can different osteotomies have an influence on surgically assisted rapid maxillary expansion? A systematic review. Head Face Med 20, 16 (2024). https://doi.org/10.1186/s13005-024-00415-3

Moreover, the risk of bias is incomplete, especially regarding the quality of the included studies. For example, one study [53] claims that MARPE is superior compared to SARPE, although the actual expansion is not shown in the work. The outcome was only ‘until crossbite correction was achieved’. This is not a comparable outcome. 

The methodology section raises an important point concerning the disproportionate inclusion of studies focused on MARPE compared to other maxillary expansion techniques such as SARPE, RPE, and dentoalveolar compensation. While the authors aim to evaluate the outcomes associated with maxillary expanders, the heavy emphasis on MARPE studies potentially undermines the objectivity and comprehensiveness of the review. The selective focus on MARPE may lead to an incomplete understanding of all available expansion techniques. 

I also miss the potential of dentoalveolar compensation (DA) (Schmid, J. Q.; Gerberding, E.; Hohoff, A.; Kleinheinz, J.; Stamm, T.; Middelberg, C. Non-Surgical Transversal Dentoalveolar Compensation with Completely Customized Lingual Appliances versus Surgically Assisted Rapid Palatal Expansion in Adults—Tipping or Translation in Posterior Crossbite Correction? https:// doi.org/10.3390/jpm13050807.)

It is understandable that a MARPE is superior to conventional RPE and DA, since MARPE, RPE and DA only affect one centre of resistance, namely the palatal suture. However, a SARPE weakens all centres of resistance in the upper jaw, which is not the case with a MARPE.

In summary, while the focus on MARPE studies might stem from contemporary interest in this methodology, the disproportionate inclusion of these studies compromises the review's ability to make objective comparisons with other established approaches. Addressing this issue in a separate section "Limitation of the study" would strengthen the paper's overall argument and provide readers with a more comprehensive understanding of maxillary expansion options. A discussion of the above mentioned articles and a more objective speech would improve the general understanding.

Author Response

Dear Reviewer,

Thank you for the your careful consideration of our scoping review. Here are the improvements we made as suggested by your comments:

Comment 1: The methodology section of this literature review appears to be well-structured and adheres broadly to the PRISMA guidelines. However, there are some critical aspects that warrant discussion, particularly concerning the scope of the literature search and the potential limitations arising from the defined 5-year review period. In particular, the investigations into SARPE are underrepresented due to the time limitation. A current systematic review could be found here: Barone, S., Bennardo, F., Salviati, M. et al. Can different osteotomies have an influence on surgically assisted rapid maxillary expansion? A systematic review. Head Face Med 20, 16 (2024). https://doi.org/10.1186/s13005-024-00415-3

Response 1: The systematic review section you suggested is now discussed in the introduction, bliography item no 9, lines 52-55.

Comment 2: Moreover, the risk of bias is incomplete, especially regarding the quality of the included studies. For example, one study [53] claims that MARPE is superior compared to SARPE, although the actual expansion is not shown in the work. The outcome was only ‘until crossbite correction was achieved’. This is not a comparable outcome. "

Response 2: The article in question, now bibliography item 54, is a cited article, published in a prestigious magazine, and was included in our review as it passed inclusion criteria and quality assessment. It provides valuable information on airway changes of MARPE compared to SARPE, with dental effects as a secondary objective. 

Comment 3: In summary, while the focus on MARPE studies might stem from contemporary interest in this methodology, the disproportionate inclusion of these studies compromises the review's ability to make objective comparisons with other established approaches. Addressing this issue in a separate section "Limitation of the study" would strengthen the paper's overall argument and provide readers with a more comprehensive understanding of maxillary expansion options. A discussion of the above mentioned articles and a more objective speech would improve the general understanding.

Response 3: This scoping review focuses on MARPE as stated in the objective: To present the recent assessments concerning MARPE influencing factors, side effects, soft tissue alterations and airway changes, also focusing on comparisons with conventional devices, therefore dental compensations are not included as choice of treatment is not discussed in this study. Our work provides valuable information on MARPE as a newer alternative to treatment with SARPE, dental compensation or conventional RPE. Also, 'Limitations of our research" has been updated to explain SARPE and RPE not being thoroughly included - lines 378-380.

Reviewer 3 Report

Comments and Suggestions for Authors

hello

thank you for this review

title matches the abstract

key words are OK

abstract is well structured

at the end of the abstract the reference to the protocol-DOI should be removed

please improve it

abstract is well presented, nothing more to change

1. chapter

introduction is sufficient

clear aim is presented

nothing to change

2. chapter

material and method section is well presented

prisma guidelines are clear

clear inclusion and exclusion criteria are written

a full description of the study is made

figure 1 is very nicely presented

all study design is good

3. chapter

results - well presented in sub paragraphs

each paragraph meets the aim of the study

nothing to change

4. results and discussion

the most important aspects are presented

both limitations and strong and week points are presenteed

nothing to change

5. end of paper

final paper consideration is good

clear study, very well presented and nicely written

I would not change anything

the amount of reviewed paper done in this PRISMA paper is sufficient

thank you

from a surgical point of view I would greatly highlight the limitations of MARPE from both surgical, anatomical and clinical point of view

minor revision

Author Response

Dear Reviewer,

Thank you for the your careful consideration of our scoping review. Here are the improvements we made as suggested by your comments:

Comments 1: at the end of the abstract the reference to the protocol-DOI should be removed"

 Response 1: DOI has been removed

Comments 2: final paper consideration is good, clear study, very well presented and nicely written, I would not change anything

Response 2: Thank you for your kind remarks.

Comments 3: from a surgical point of view I would greatly highlight the limitations of MARPE from both surgical, anatomical and clinical point of view

Response 3: Lines 52-55 have been updated by including a new bibliography item. also, in "limitations of this research" lines 378-380 have been added.

Round 2

Reviewer 1 Report

Comments and Suggestions for Authors

The authors responded to all comments and made the required corrections.

Reviewer 2 Report

Comments and Suggestions for Authors

The authors have attempted to provide a more objective account with the revision.